# Impact of horizontal gene transfer on emergence and stability of cooperative virulence in *Salmonella* Typhimurium

Erik Bakkeren [1,6,7], Ersin Gül [1], Jana S. Huisman [2,3], Yves Steiger[1], Andrea Rocker[4], Wolf-Dietrich Hardt [1,8✉] & Médéric Diard [4,5,8✉]

Intestinal inflammation fuels the transmission of *Salmonella* Typhimurium (*S*.Tm). However, a substantial fitness cost is associated with virulence expression. Mutations inactivating transcriptional virulence regulators generate attenuated variants profiting from inflammation without enduring virulence cost. Such variants interfere with the transmission of fully virulent clones. Horizontal transfer of functional regulatory genes (HGT) into attenuated variants could nevertheless favor virulence evolution. To address this hypothesis, we cloned *hilD*, coding for the master regulator of virulence, into a conjugative plasmid that is highly transferrable during intestinal colonization. The resulting mobile *hilD* allele allows virulence to emerge from avirulent populations, and to be restored in attenuated mutants competing against virulent clones within-host. However, mutations inactivating the mobile *hilD* allele quickly arise. The stability of virulence mediated by HGT is strongly limited by its cost, which depends on the *hilD* expression level, and by the timing of transmission. We conclude that robust evolution of costly virulence expression requires additional selective forces such as narrow population bottlenecks during transmission.

[1] Institute of Microbiology, Department of Biology, ETH Zurich, Zurich, Switzerland. [2] Institute of Integrative Biology, Department of Environmental Systems Science, ETH Zurich, Zurich, Switzerland. [3] Swiss Institute of Bioinformatics, Lausanne, Switzerland. [4] Biozentrum, University of Basel, Basel, Switzerland. [5] Botnar Research Centre for Child Health, Basel, Switzerland. [6] Present address: Department of Zoology, University of Oxford, Oxford, UK. [7] Present address: Department of Biochemistry, University of Oxford, Oxford, UK. [8] These authors jointly supervised this work: Wolf-Dietrich Hardt, Médéric Diard. ✉email: wolf-dietrich.hardt@micro.biol.ethz.ch; mederic.diard@unibas.ch

**B**acteria often exist in dense communities. Therefore, many aspects of bacterial lifestyle are governed by social interactions[1]. This has also been observed for pathogens, which often infect hosts through collective actions[2–4]. For example, they can secrete extracellular metabolites or enzymes to assist in growth (e.g. iron-scavenging siderophores)[3], produce toxins to compete with other species[5,6], establish and survive within biofilms[7], or use virulence factors to modulate the host immune response to create a favorable environment[8]. These collective actions function through public goods, which are costly to produce. Hence, cheater mutants can emerge by profiting from the public good without enduring the cost of its production. In extreme cases, the overgrowth by cheaters can lead to population collapse due to the total breakdown of public good production[2]. Gene regulation, phenotypic heterogeneity, population structure, and ecological factors[9,10] likely contribute, if not to their emergence, at least to the stability of cooperative traits by altering the cost/benefit ratio[2,7,11–13].

The enteric pathogen *Salmonella enterica* serovar Typhimurium (*S*.Tm) favors its own growth, but also that of related *Enterobacteriaceae* by actively triggering gut inflammation via its Type Three Secretion System-1 (TTSS-1) and secreted effectors[8,14]. Production of TTSS-1 and co-regulated functions controlled by HilD[15–18] is associated with a cost both in vitro[19] and in vivo[11], which favors the emergence of attenuated cheaters during infection[11]. The virulence of *S*.Tm is therefore a cooperative trait. The target of selection is the transcriptional regulation of virulence expression via HilD[11]. Mutants in *hilD* have been isolated from patients[20] and swine[21], and are under niche-specific positive selection according to comparative genomic analyses of more than 100,000 natural isolates of *Salmonella enterica*[22]. When co-transmitted with virulent clones, high frequencies of cheaters prevent the disease in recipient hosts[23]. This suggests that artificial introduction of cheaters in a population of hosts could be a viable biocontrol strategy against *Salmonella* spp. However, horizontal gene transfer (HGT), known to accelerate the evolution of virulence in pathogenic bacteria[4], could potentially restore virulence in cheaters[24,25]. Nevertheless, over time, cheating should re-occur by mutation at the level of the vector[26], in the same way that mutations in *hilD* occur on the chromosome turning cooperators into cheaters[11,22]. Here, we show that this prediction is correct. Using the mouse model to study the emergence of cooperative (i.e., cheatable) virulence and its stability in *S*.Tm during intestinal colonization and transmission, we demonstrate the rise and fall of cooperative virulence mediated by HGT during within-host evolution. This context captures the complexity of the host-pathogen interaction in which cheaters can naturally evolve[11,27]. We hypothesized that although HGT may favor cooperative virulence, host-to-host transmission timing and associated population bottlenecks should stabilize cooperative virulence in the long run, since virulent clones trigger disease and promote shedding whereas cheating clones do not[23].

## Results

### Evolution of cooperative virulence in an avirulent population mediated by HGT in vivo

We created a tractable model to address the role of HGT in the evolution of cooperative virulence by cloning *hilD* coupled to a chloramphenicol resistance cassette into the conjugative *IncI*1 plasmid P2 (aka. pCol1B9), which is native to *S*.Tm SL1344[14]. The resulting construct was named pVir. We have previously shown that P2 spreads efficiently into *S*.Tm 14028S and some *E. coli* strains in vivo[14,28–31]. This system allows robust HGT independently of intestinal inflammation, which is not the case for other vectors such as temperate bacteriophages like SopEΦ[30]. As a donor strain, we conjugated pVir

into an *S*.Tm 14028S derivative that lacks both a chromosomal copy of *hilD* and a functional SPI-1 locus (Fig. 1a and Table 1). As a recipient strain, we used a kanamycin-resistant derivative of *S*.Tm 14028S that also lacks a chromosomal copy of *hilD*, but has all necessary genes to produce a functional TTSS-1 (Fig. 1a). Both the donor and the recipient are genetically avirulent (i.e. they do not elicit overt gut inflammation), but conjugation of pVir to the recipient should produce a transconjugant able to trigger inflammation (Fig. 1a). Both the donor and the recipient lack a functional TTSS-2 (*ssaV* mutant) to exclude inflammation triggered through TTSS-2 at a later stage in mouse gut infections[32,33].

To determine to which extent cooperative virulence evolution depends on the fitness cost of the cooperative allele[34], we constructed two variants of pVir. The "low cost" variant (pVir^Low) contains 648 bp of the regulatory region upstream of *hilD* (and all four transcriptional start sites characterized in Kroger et al.[15]), while the "high cost" variant (pVir^High) contains 279 bp of upstream regulatory region (only two transcriptional start sites of *hilD*;[15] Fig. S1A). To test the difference in cost associated with each pVir variant, we performed in vitro experiments comparing growth rate with TTSS-1 expression, which is associated with a cost[19] and thus inversely correlated to growth (Fig. S1B, C). We confirmed that transconjugants harboring pVir^Low expressed less TTSS-1 and grew better than those harboring pVir^High, confirming the difference in cost associated with TTSS-1 expression induced by these constructions (Fig. S1B, C).

To address the conditions that may support the evolution of cooperative virulence, we performed conjugation experiments in an antibiotic pretreated mouse model (modified from Barthel et al.[35]). We introduced the donor and recipient strains sequentially into ampicillin pretreated mice at low inoculum size ($10^2$ CFU donors; $10^4$ CFU recipients) to ensure that conjugation occurred only in the gut. By 2 days post infection, 97% of recipients (median of all mice) obtained the plasmid, although spread of pVir^Low and pVir^High proceeded slower than the P2 control plasmid (pVir lacking *hilD*; i.e. P2^*cat*; Fig. 1b, day 1 p.i.). The plasmid was also maintained in the majority of mice for the entire course of the experiment with a median of 45% recipient cells carrying the plasmid. To test if transfer of pVir could allow the emergence of cooperative virulence in a population of avirulent recipients, we measured fecal lipocalin-2 (LCN2) as a readout for the inflammatory status of the gut. Inflammation was progressively triggered as more pVir transconjugants were formed, leading to a maximum at 3 days post infection (Fig. 1b, c). Mice containing *S*.Tm with either pVir^Low or pVir^High were significantly more inflamed than mice infected with control *S*.Tm donors (day 2–7 post infection; Fig. 1d). This shows that virulence can evolve within a host, since neither the donor, nor the recipient were virulent prior to conjugation (Fig. S2). However, intestinal inflammation was not sustained (Fig. 1c), and mice began to recover leading to exclusion of *S*.Tm from the gut likely by the re-growing microbiota (Figs. 1d and S3)[36]. Furthermore, mice harboring virulent *S*.Tm did not excrete significantly more *S*.Tm than control mice (Fig. 1d and Supplementary discussion). This observation led to two questions: (1) what makes emerged cooperative virulence short lived, and (2) can HGT of a cooperative allele favor the transmission of the disease?

### HGT-mediated cooperative virulence is short lived, characterized by cost-dependent inactivation of the mobile cooperative allele

We hypothesized that the waning inflammation was a result of insufficient TTSS-1 expression. Inflammation started to decrease after day 3 post infection, while the proportion of

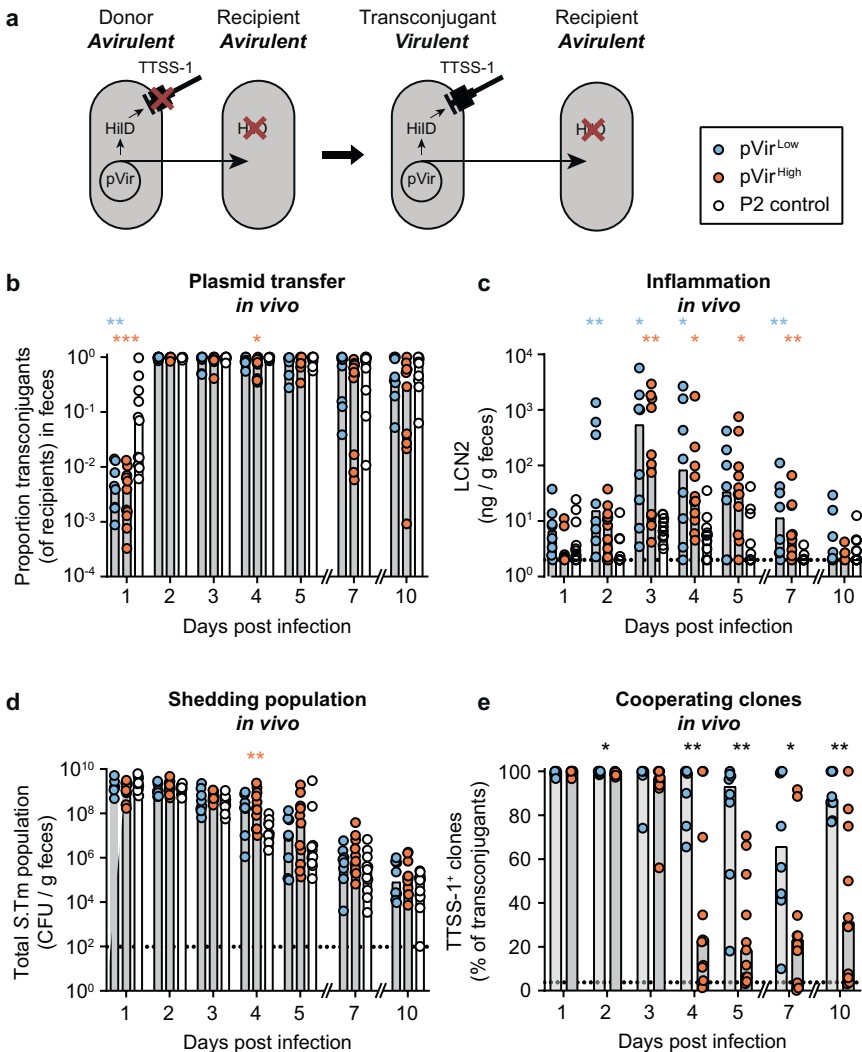

**Fig. 1 Virulence can emerge through HGT in a population of cheaters in vivo, but is unstable. a** Experimental system to measure maintenance of cooperative virulence by HGT. Donors contain pVir encoding *hilD*, but cannot produce a functional TTSS-1 (*invG* mutant), making them avirulent. Recipients contain all genes for a functional TTSS-1 but do not have a functional copy of *hilD* (cheaters), preventing TTSS-1 expression and virulence. Transfer of pVir from the donor to the recipient forms a transconjugant that contains both functional TTSS-1-encoding genes and a copy of *hilD* from pVir, allowing TTSS-1-mediated virulence. Transconjugants can transfer pVir to additional recipients. **b–e** pVir is transferred to cheater recipients and allows cooperative virulence to emerge. Ampicillin pretreated mice were sequentially infected orally with donors (14028S Δ*invG* Δ*hilD* Δ*ssaV*; Cm^R, Amp^R) harboring pVir^Low (blue; *n* = 8), pVir^High (orange; *n* = 11), or P2 lacking *hilD* (control; white; *n* = 11), and recipients (14028 S Δ*hilD* Δ*ssaV*; Kan^R, Amp^R). Each replicate is shown and bars indicate the median. Source data are provided as a Source Data file. **b–d** Statistics compare pVir^Low (blue asterisks) and pVir^High (orange asterisks) to the control on each day; Kruskal–Wallis test with Dunn's multiple test correction ($p > 0.05$ not significant and not indicated, *$p < 0.05$, **$p < 0.01$, ***$p < 0.001$. Dotted line represents the detection limit. **b** Plasmid transfer was measured by selective and/or replica plating. The proportion of transconjugants is calculated by dividing the transconjugant population by the sum of recipients and transconjugants. **c** Inflammation was measured by a Lipocalin-2 ELISA on fecal samples. **d** Total population determined by summing all subpopulations. Donor, recipient, and transconjugant populations are presented in Fig. S3. **e** Transconjugants carrying pVir^Low (blue) and pVir^High (orange) were analyzed by colony western-blot and compared using a two-tailed Mann–Whitney *U* test ($p > 0.05$ not significant and not indicated, *$p < 0.05$, **$p < 0.01$, ***$p < 0.001$, ****$p < 0.0001$). The percentage of colonies that expressed SipC are reported out of the total transconjugant population. Bars indicate the median. The black dotted line indicates the conservative detection limit, which is dependent on the number of colonies on the plate (values can therefore appear below the detection limit).

transconjugants in the feces of mice infected with pVir-harboring *S*.Tm remained 84% at day 4 post infection (median of all mice with pVir^High or pVir^Low; Fig. 1b). Therefore, plasmid loss could not explain cooperative virulence loss. However, cheating could have occurred by mutations on the plasmid, as previously suggested[25,26].

To address this, we performed a western blot on transconjugant colonies isolated from feces ("colony blot")[11,27] to probe for TTSS-1 expression. As expected, cooperative virulence emergence by HGT was transient, since clones that do not express TTSS-1 arose

(Fig. 1e). We performed whole-genome sequencing on evolved clones that were either TTSS-1^+ or TTSS-1^− as determined by the colony blot (*n* = 11 for pVir^Low; *n* = 14 clones for pVir^High). It showed that cheating was a result of mutations or deletions in either the coding sequence or regulatory regions of *hilD* on pVir, and not due to chromosomal changes (Tables S1–4). HGT-mediated cooperation is therefore short lived, since cheating now occurred at the level of the mobile genetic element (MGE). As expected, the loss of TTSS-1^+ clones was slower with pVir^Low compared to pVir^High, leading to a higher proportion of cooperating clones

**Table 1 Strains used in this study.**

| Strain name | Strain number | Relevant genotype[a] | Resistance[b] | Reference |
|---|---|---|---|---|
| SL1344 | SB300 | Wild type | Sm | [65] |
| ATCC 14028S | 14028S | Wild type | None | [66] |
| *E. coli* CC118 λpir | – | λpir; used for R6K ori replication (e.g. in pKD3) | None | [67,68] |
| SB300 ΔhilD | M3101 | ΔhilD | Sm | [11] |
| SB300 ΔhilD pVir$^{Low}$ | Z2325 | ΔhilD pVir$^{Low}$ | Sm, Cm | This work |
| SB300 ΔhilD pVir$^{High}$ | Z2326 | ΔhilD pVir$^{High}$ | Sm, Cm | This work |
| 14028S ΔhilD ΔinvG ΔssaV pM975 | Z2327 | 14028S ΔhilD ΔinvG ΔssaV pM975 | Amp | This work |
| Low cost pVir donor | Z2317 | 14028S ΔhilD ΔinvG ΔssaV pVir$^{Low}$ pM975 | Amp, Cm | This work |
| High cost pVir donor | Z2236 | 14028S ΔhilD ΔinvG ΔssaV pVir$^{High}$ pM975 | Amp, Cm | This work |
| Control donor | Z2318 | 14028S ΔhilD ΔinvG ΔssaV P2$^{cat}$ pM975 | Amp, Cm | This work |
| Cheater recipient | Z2235 | 14028S ΔhilD ssaV::aphT pM975 | Amp, Kan | This work |
| Wild-type pM972 | Z2319 | 14028S ssaV::aphT pM972 | Amp, Kan | This work |
| ΔhilD recipient pM972 | Z2320 | 14028S ΔhilD ssaV::aphT pM972 | Amp, Kan | This work |
| 14028S ΔhilD ΔssaV pM975 | T144 | 14028S ΔhilD ΔssaV pM975 | Amp | This work |
| 14028S ΔhilD ΔssaV pM975 pVir$^{Low}$ | T1 | 14028S ΔhilD ΔssaV pVir$^{Low}$ pM975 | Amp, Cm | This work |
| Cheater recipient P2 | T154 | 14028S ΔhilD ΔssaV P2$^{aphT}$ pM975 | Amp, Kan | This work |
| ΔhilD pVir$^{Low}$ transconjugant pM972 −1 | Z2321 | 14028S ΔhilD ssaV::aphT pM972 pVir$^{Low}$ | Amp, Kan, Cm | This work |
| ΔhilD pVir$^{Low}$ transconjugant pM972 −2 | Z2322 | 14028S ΔhilD ssaV::aphT pM972 pVir$^{Low}$ | Amp, Kan, Cm | This work |
| ΔhilD pVir$^{High}$ transconjugant pM972 −1 | Z2323 | 14028S ΔhilD ssaV::aphT pM972 pVir$^{High}$ | Amp, Kan, Cm | This work |
| ΔhilD pVir$^{High}$ transconjugant pM972 −3 | T292 | 14028S ΔhilD ssaV::aphT pM972 pVir$^{High}$ | Amp, Kan, Cm | This work |
| ΔhilD pVir$^{High}$ transconjugant pM972 −4 | T293 | 14028S ΔhilD ssaV::aphT pM972 pVir$^{High}$ | Amp, Kan, Cm | This work |
| ΔhilD pVir$^{High}$ transconjugant pM972 −5 | T294 | 14028S ΔhilD ssaV::aphT pM972 pVir$^{High}$ | Amp, Kan, Cm | This work |
| ΔhilD pVir$^{High}$ transconjugant pM972 −6 | T295 | 14028S ΔhilD ssaV::aphT pM972 pVir$^{High}$ | Amp, Kan, Cm | This work |
| ΔhilD pVir$^{High}$ transconjugant pM972 −7 | T296 | 14028S ΔhilD ssaV::aphT pM972 pVir$^{High}$ | Amp, Kan, Cm | This work |
| Evolved transconjugant Low cost TTSS-1$^+$ −1 | Z2296 | 14028S ΔhilD ssaV::aphT pM975 pVir$^{Low}$ | Amp, Kan, Cm | This work |
| Evolved transconjugant Low cost TTSS-1$^+$ −2 | Z2306 | 14028S ΔhilD ssaV::aphT pM975 pVir$^{Low}$ | Amp, Kan, Cm | This work |
| Evolved transconjugant Low cost TTSS-1$^+$ −3 | Z2310 | 14028S ΔhilD ssaV::aphT pM975 pVir$^{Low}$ | Amp, Kan, Cm | This work |
| Evolved transconjugant Low cost TTSS-1$^+$ −4 | Z2299 | 14028S ΔhilD ssaV::aphT pM975 pVir$^{Low}$ | Amp, Kan, Cm | This work |
| Evolved transconjugant Low cost TTSS-1$^+$ −5 | Z2302 | 14028S ΔhilD ssaV::aphT pM975 pVir$^{Low}$ | Amp, Kan, Cm | This work |
| Evolved transconjugant Low cost TTSS-1$^+$ −6 | Z2308 | 14028S ΔhilD ssaV::aphT pM975 pVir$^{Low}$ | Amp, Kan, Cm | This work |
| Evolved transconjugant Low cost TTSS-1$^-$ −1 | Z2298 | 14028S ΔhilD ssaV::aphT pM975 pVir$^{Low}$ | Amp, Kan, Cm | This work |
| Evolved transconjugant Low cost TTSS-1$^-$ −2 | Z2305 | 14028S ΔhilD ssaV::aphT pM975 pVir$^{Low}$ | Amp, Kan, Cm | This work |
| Evolved transconjugant Low cost TTSS-1$^-$ −3 | Z2301 | 14028S ΔhilD ssaV::aphT pM975 pVir$^{Low}$ | Amp, Kan, Cm | This work |
| Evolved transconjugant Low cost TTSS-1$^-$ −4 | Z2304 | 14028S ΔhilD ssaV::aphT pM975 pVir$^{Low}$ | Amp, Kan, Cm | This work |
| Evolved transconjugant Low cost TTSS-1$^-$ −5 | Z2309 | 14028S ΔhilD ssaV::aphT pM975 pVir$^{Low}$ | Amp, Kan, Cm | This work |
| Evolved transconjugant High cost TTSS-1$^+$ −1 | Z2238 | 14028S ΔhilD ssaV::aphT pM975 pVir$^{High}$ | Amp, Kan, Cm | This work |
| Evolved transconjugant High cost TTSS-1$^+$ −2 | Z2253 | 14028S ΔhilD ssaV::aphT pM975 pVir$^{High}$ | Amp, Kan, Cm | This work |
| Evolved transconjugant High cost TTSS-1$^+$ −3 | Z2246 | 14028S ΔhilD ssaV::aphT pM975 pVir$^{High}$ | Amp, Kan, Cm | This work |
| Evolved transconjugant High cost TTSS-1$^+$ −4 | Z2242 | 14028S ΔhilD ssaV::aphT pM975 pVir$^{High}$ | Amp, Kan, Cm | This work |
| Evolved transconjugant High cost TTSS-1$^+$ −5 | Z2244 | 14028S ΔhilD ssaV::aphT pM975 pVir$^{High}$ | Amp, Kan, Cm | This work |
| Evolved transconjugant High cost TTSS-1$^+$ −6 | Z2312 | 14028S ΔhilD ssaV::aphT pM975 pVir$^{High}$ | Amp, Kan, Cm | This work |
| Evolved transconjugant High cost TTSS-1$^-$ −1 | Z2239 | 14028S ΔhilD ssaV::aphT pM975 pVir$^{High}$ | Amp, Kan, Cm | This work |
| Evolved transconjugant High cost TTSS-1$^-$ −2 | Z2243 | 14028S ΔhilD ssaV::aphT pM975 pVir$^{High}$ | Amp, Kan, Cm | This work |
| Evolved transconjugant High cost TTSS-1$^-$ −3 | Z2311 | 14028S ΔhilD ssaV::aphT pM975 pVir$^{High}$ | Amp, Kan, Cm | This work |
| Evolved transconjugant High cost TTSS-1$^-$ −4 | Z2245 | 14028S ΔhilD ssaV::aphT pM975 pVir$^{High}$ | Sm, Kan, Cm | This work |
| Evolved transconjugant High cost TTSS-1$^-$ −5 | Z2247 | 14028S ΔhilD ssaV::aphT pM975 pVir$^{High}$ | Amp, Kan, Cm | This work |
| Evolved transconjugant High cost TTSS-1$^-$ −6 | Z2252 | 14028S ΔhilD ssaV::aphT pM975 pVir$^{High}$ | Amp, Kan, Cm | This work |
| Evolved transconjugant High cost TTSS-1$^-$ −7 | Z2254 | 14028S ΔhilD ssaV::aphT pM975 pVir$^{High}$ | Amp, Kan, Cm | This work |
| Evolved transconjugant High cost TTSS-1$^-$ −8 | Z2255 | 14028S ΔhilD ssaV::aphT pM975 pVir$^{High}$ | Amp, Kan, Cm | This work |

[a]For additional information on genotypes of select strains, see Tables S1–4 for a whole-genome resequencing summary.
[b]Relevant resistances only: Sm = ≥50 µg/ml streptomycin; Cm = ≥15 µg/ml chloramphenicol; Kan = ≥50 µg/ml kanamycin; Amp = ≥100 µg/ml ampicillin.

bearing pVir$^{Low}$ by the end of the experiment (Fig. 1e). This indicates that cost influences the maintenance of cooperative virulence mediated by HGT (as predicted by theory[24,26]), extending our previous in vivo work on the cost-dependent cheating dynamics of S.Tm with the chromosomal copy of hilD[11].

However, since both the cheating dynamics and the total population size dictate the size of the population able to trigger inflammation, we multiplied the proportion of transconjugants able to trigger inflammation (from Fig. 1e) by the transconjugant population size (Fig. S3A, B) to obtain the effective size of the cooperative population (i.e., the TTSS-1$^+$ population; Fig. S3D). We observed a rise in the cooperative population due to plasmid transfer correlating with the onset of inflammation (day 1–2 p.i.,

Fig. S3D), followed by a drop in the cooperative population associated with the waning inflammation between days 5–10 p.i., Fig. S3D). This supported the hypothesis that the loss of inflammation could be driven by the loss of the cooperative population. As predicted by analyzing the proportion of transconjugants able to produce TTSS-1 (Fig. 1e), the cooperative population was higher in mice infected with pVir$^{Low}$ harboring cells compared to those infected with pVir$^{High}$ harboring cells at the end of the experiment (Day 10; Fig. S3D).

**The cost of virulence expression drives the proportion of cooperators shed over generations.** In our model system, HGT allowed cooperative virulence to emerge, however it remains

unstable within-host. In the case of S.Tm, TTSS-1-triggered inflammation has two important consequences that can nevertheless promote cooperative virulence: it favors transmission of S.Tm[8,37] and fosters pathogen blooms in the next host. Therefore, HGT may influence the evolution of S.Tm by increasing the duration of shedding of sufficient virulent clones to colonize and to trigger inflammation in a new host. Selection for cooperative virulence should be a result of increased benefit after transmission. To address this hypothesis, we took fecal suspensions from mice in Fig. 1 on day 2 p.i. (the maximum population size of TTSS-1-producing clones; Fig. S3D) and day 10 p.i. (the minimum population size of TTSS-1-producing clones; Fig. S3D) and transferred them into new ampicillin pretreated mice (Fig. 2a).

When fecal populations taken from day 2 p.i. were transferred into new mice, inflammation was triggered, and there were no significant differences between mice infected with pVir$^{Low}$ or pVir$^{High}$ S.Tm carriers (Fig. 2b). This led to consistent shedding over the course of the experiment in all the recipient mice (Figs. 2c and S4A–C). This is likely attributed to the high proportion of cooperating clones transferred with samples from day 2 p.i. (Fig. 1). However, in all mice, cheaters arose (Fig. 2d), further supporting the instability of cooperative virulence in this system. In contrast, when feces from day 10 p.i. were transferred, only the recipient mice infected with S.Tm harboring pVir$^{Low}$ became inflamed (Fig. 2e). This was reflected in the shedding population at day 4 post transmission, where mice infected with S.Tm harboring pVir$^{Low}$ contained significantly more S.Tm in the feces compared to mice infected with S.Tm containing pVir$^{High}$ (Figs. 2f and S4D–F). These differences were likely a result of the proportion of cooperative clones in the feces of the donor mice at day 10 p.i.: mice infected with S.Tm pVir$^{Low}$ had significantly more cooperators than mice infected with S.Tm pVir$^{High}$ (Figs. 1 and S3). Moreover, as in the fecal transfer at day 2 p.i., in all mice, cheaters arose and outgrew cooperators (Fig. 2g). Interestingly, three mice with S.Tm pVir$^{High}$ that contained a high proportion of cooperative clones at day 10 p.i. (Fig. 1e) did not lead to strong inflammation after transmission (Fig. 2e). This could indicate that additional cost-dependent factors influence the ability to trigger inflammation after transmission. Nevertheless, since transmission of feces containing pVir$^{High}$ S.Tm led to inflammation dependent on the proportion of cooperators (e.g. compare the fecal transfer at day 2 p.i. (Fig. 2b) to the transfer at day 10 p.i. (Fig. 2e)), we concluded that the proportion of cooperators do contribute to disease development post transmission.

Altogether, this confirms that triggering disease in the next host requires sufficient proportion of cooperators in the transmitted population. This is in accordance with previous observations that a population made of 99% cheaters cannot trigger the disease after transmission[23] and that at least 10 to 50% of the gut luminal S.Tm population must encode a functional TTSS-1 to provoke host response[38]. The proportion of cheaters depends mainly on the cost of virulence expression and on the duration of evolution within the donor. Long-term colonization and high cost are detrimental to stability of virulence as cheaters are more likely to reach high frequency before transmission. Moreover, co-transmitted cheaters keep accumulating in the new host, eventually reaching fixation, which prevents disease and further transmission.

**Genetic drift favors cooperative virulence.** In our system, HGT can allow the re-emergence of cooperating clones. However, cooperation remained unstable within-host as mutations inactivating the cooperative allele on the plasmid arose quickly. Narrow transmission bottlenecks, that is, when few founding members lead to the establishment of new populations[10], could nevertheless promote the stability of cooperative virulence by purging cheaters from the population. A population bottleneck at transmission can be the result of environmental stress between hosts, dilution of the bacterial population in the environment (followed by growth to reach the infective dose, e.g. in contaminated foods) or due to competition against the resident microbiota before establishing a favorable niche in the gut[39]. To address this, we performed a proof of principle experiment simulating an extreme population bottleneck by infecting new mice with single evolved clones from mice in Fig. 1. We used clones evolved in different mice, representing both cooperators (i.e., TTSS-1$^+$ clones) and cheaters (i.e., TTSS-1$^-$ clones) in both the high cost and low cost variants (3 clones per group; 2-3 new mice infected per clone). For both pVir$^{Low}$ and pVir$^{High}$, TTSS-1$^+$ clones were able to trigger inflammation (LCN2 ≥ 10$^2$ ng/g feces) and TTSS-1$^-$ clones were not (Fig. 3a). Again, this was reflected in the shedding population, where mice infected with TTSS-1$^+$ clones shed significantly more S.Tm on day 4 p.i. compared to mice infected with TTSS-1$^-$ clones (Fig. 3b). Note that the antibiotic pretreatment allows pure cheater populations to reach the same population size as cooperators for three days post-infection. As a control, we measured the proportion of cooperative clones in these mice on day 4 p.i. As expected, mice infected with cheater clones contained only TTSS-1$^-$ clones and mice infected with cooperative clones contained mostly TTSS-1$^+$ clones, although cheaters began to emerge in these mice as well (Fig. 3c). Importantly, the proportion of cooperators in mice infected with single TTSS-1$^+$ clones appeared higher than in mice that received a non-isogenic mixture of cooperators and cheaters (compare Fig. 3c to 2d, g). In accordance with theory and experimental works demonstrating that assortment favors cooperative traits[3,40], monoclonal infections clearly promote cooperative virulence. To further support this point, we tested the correlation between the proportion of cooperators given to mice in the transmission experiments (Figs. 2 and 3) and inflammation, the shedding population, and the proportion of cooperators at day 4 post infection (Fig. S5). The proportion of cooperators in the input correlated with the resulting inflammation, shedding, and the final proportion of cooperative clones (slopes are significantly non-zero: Fig. S5A $p < 0.0001$; Fig. S5B $p = 0.0002$; Fig. S5C $p < 0.0001$), in line with previous work comparing the proportion of the population able to express the TTSS-1$^+$ phenotype and the resulting inflammation[38].

**HGT can maintain cooperation in a virulent population invaded by cheaters.** After addressing the emergence of cooperative virulence in an avirulent population and the role of transmission in its maintenance, we assessed the contribution of HGT in maintaining cooperation within-host in a population of cooperators competing against cheaters (as proposed in Smith[24]). In this experiment, the donor contained a functional invG allele, making it virulent (in comparison to the scenario in Fig. 1 where the donor is avirulent). We performed a competitive infection between a hilD mutant (i.e., cheater) and the donor containing pVir$^{Low}$ (i.e., cooperator) at an equal ratio (low inoculum size; ~10$^2$ CFU of each strain, introduced sequentially to avoid plasmid transfer in the inoculum). Importantly, we performed this experiment in two configurations (Fig. 4a): in the first group which we called "mobile pVir", we used the same recipient cheater strain as in Fig. 1, which can obtain pVir from the cooperator; in the second group called "non-mobile pVir", we used a cheater strain that carried P2 (P2 does not confer a fitness advantage to competing 14028 S strains[14,31]) labeled with a kanamycin resistance marker. In this case, since pVir and P2 are incompatible and have mechanisms for entry exclusion[41],

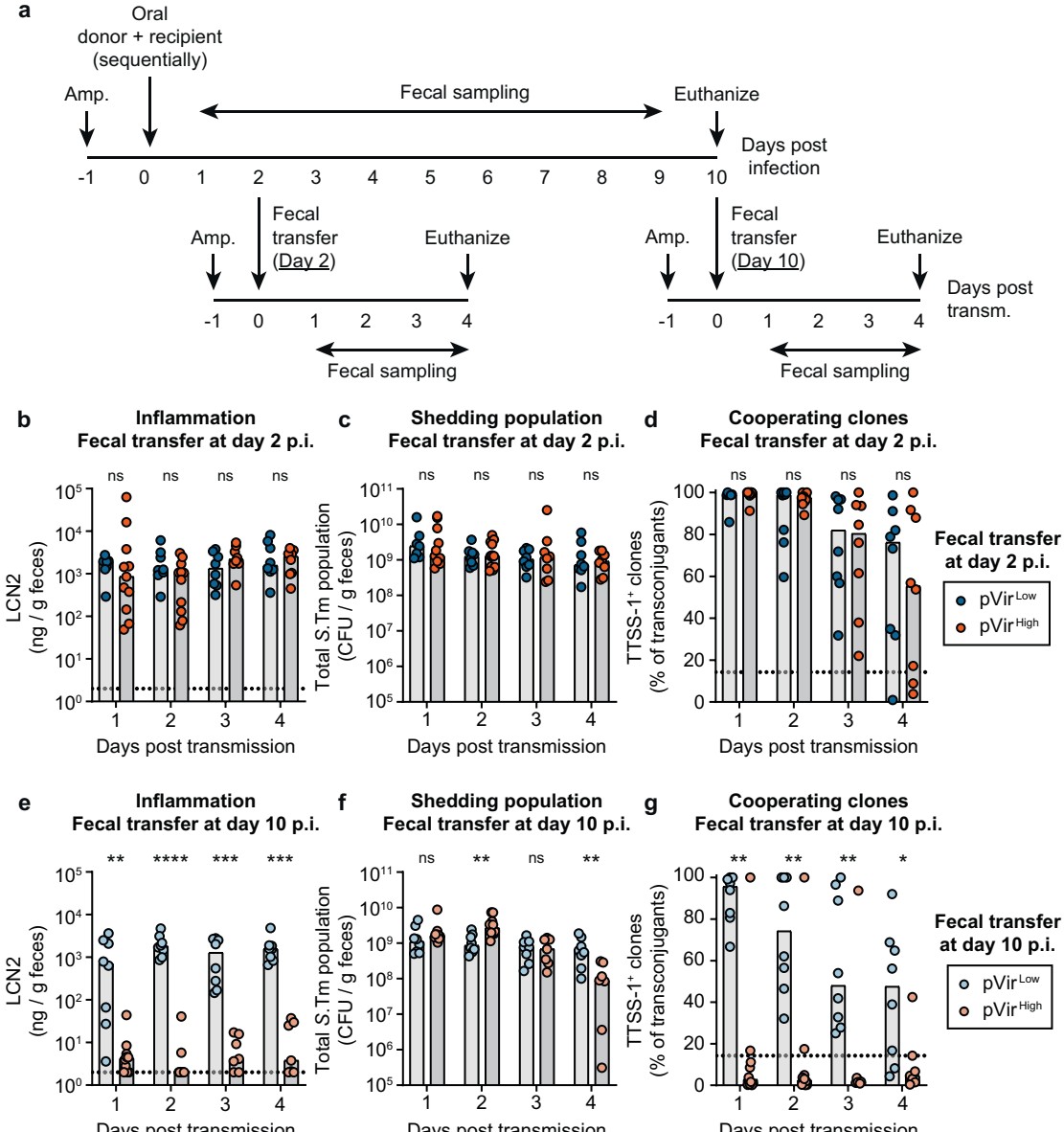

**Fig. 2 Successful infections in new hosts depend on the proportion of transmitted cooperators. a** Experimental scheme for transmission experiments. Feces from mice in Fig. 1 collected on day 2 and day 10 post infection were suspended in PBS and given to new ampicillin (Amp.) pretreated mice. **b–g** Mice orally given fecal resuspensions with S.Tm harboring pVir^Low (blue; dark shade for day 2 transmission; light shade for day 10 transmission; n = 8 for both groups) are compared to pVir^High (orange; dark shade for day 2 transmission (n = 11); light shade for day 10 transmission (n = 10)) using a two-tailed Mann–Whitney U test (p > 0.05 (ns), *p < 0.05, **p < 0.01, ***p < 0.001, ****p < 0.0001). All data points are shown and medians are represented by bars. Source data are provided as a Source Data file. **b–d** Mice given feces from day 2 post infection. **e–g** Mice given feces from day 10 post infection. **b**, **e** Inflammation was quantified using a LCN2 ELISA. The dotted lines indicate the detection limit. **c**, **f** The shedding population was enumerated by summing all populations determined by selective plating. Donor, recipient, and transconjugant populations are presented in Fig. S4. **d**, **g** MacConkey plates containing colonies of transconjugants were analyzed for expression of SipC as a proxy for TTSS-1 expression using a colony western blot; the percentage of colonies that expressed SipC are reported out of the total transconjugant population. The black dotted line indicates the conservative detection limit for the colony blot, which is dependent on the number of colonies on the plate (values can therefore appear below the detection limit).

plasmid transfer cannot be detected (confirmed by selective plating). As expected, when pVir was mobile, the plasmid was maintained in the population for longer compared to the non-mobile scenario, in which the cooperating strain was out-competed by the cheating strain (Figs. 4b, c and S6). Furthermore, inflammation was maintained for longer in mice with the mobile pVir scenario (Fig. 4d), which was reflected in a trend towards higher shedding populations (Fig. 4e). Importantly, in some mice with the mobile pVir scenario, the inflammation and the shedding population also diminished over time (Fig. 4d, e). Therefore,

we measured the proportion of TTSS-1-expressing clones in the population at the end of the experiment. Although more TTSS-1-expressing clones were observed in the mobile pVir scenario compared to when pVir was not mobile (Fig. 4f), clones that did not express TTSS-1 were detected within the pVir-containing population (Fig. 4g).

## Discussion
Using an experimental evolution model in vivo we show that HGT can facilitate restoration, and possibly the emergence of

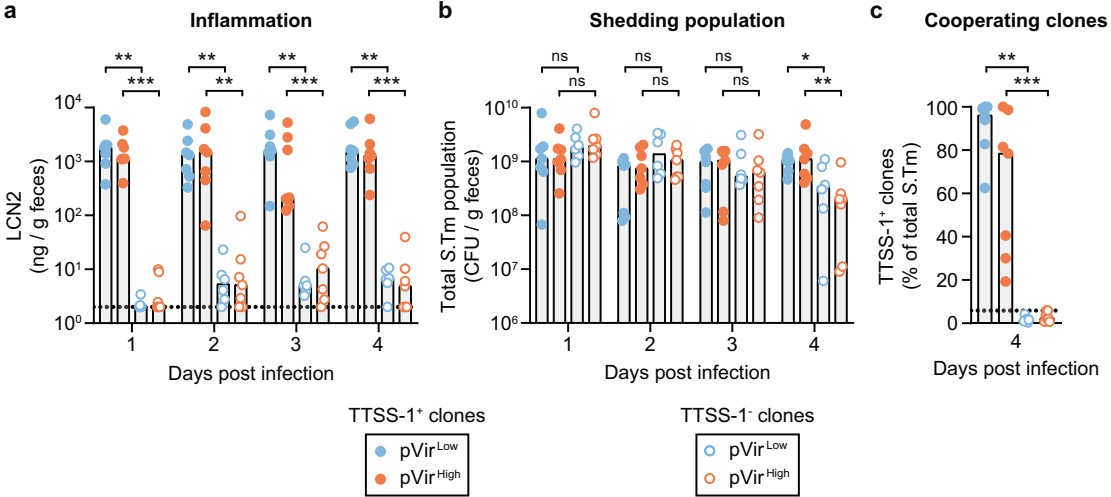

**Fig. 3 Genetic drift favors cooperative virulence.** Ampicillin pretreated mice were orally infected with evolved transconjugant clones isolated from day 7 or day 10 from mice in Fig. 1. Three cooperating clones (solid circles; TTSS-1[+] clones) and cheating clones (hollow circles; TTSS-1[−] clones) were randomly chosen for each of pVir[Low] (blue) or pVir[High] (orange). Each clone was infected into 2–3 mice (~5 × 10[7] CFU inoculum), leading to a total of 6–7 mice per group. All clones were whole-genome sequenced (mutations and indels are summarized in Tables S1–4): TTSS-1[+] pVir[Low] (Z2296 (3 mice), Z2306 (2 mice), Z2310 (2 mice); $n = 7$), TTSS-1[+] pVir[High] (Z2238 (3 mice), Z2246 (2 mice), Z2253 (2 mice); $n = 7$), TTSS-1[−] pVir[Low] (Z2298, Z2301, Z2305; 2 mice per clone; $n = 6$), TTSS-1[−] pVir[High] (Z2239 (3 mice), Z2243 (2 mice), Z2311 (2 mice); $n = 7$). All data points are shown and medians are indicated by bars. Comparisons are made between TTSS-1[+] and TTSS-1[−] clones (for each of pVir[Low] and pVir[High]) using a two-tailed Mann–Whitney U test ($p > 0.05$ (ns), *$p < 0.05$, **$p < 0.01$, ***$p < 0.001$). Source data are provided as a Source Data file. **a** Inflammation was quantified using a LCN2 ELISA. The dotted lines indicate the detection limit. **b** The shedding population was enumerated on MacConkey agar. **c** MacConkey plates containing colonies were analyzed for expression of SipC as a proxy for TTSS-1 expression using a colony western blot; the percentage of colonies that expressed SipC are reported out of the total transconjugant population. The black dotted line indicates the conservative detection limit for the colony blot, which is dependent on the number of colonies on the plate (values can therefore appear below the detection limit).

cooperative virulence. However, we observed that cheating re-occurred quickly in recipient bacteria via mutations inactivating the mobile cooperative allele (Tables S1–S4). Because both the mobile cooperative and mutated "cheating" alleles spread equally well in the population, here via conjugation, cooperation is only restored transiently. This has been previously shown in vitro and predicted from several theoretical scenarios[25,26,42,43]. Moreover, the concept that cooperative alleles are more often encoded on MGE because HGT could maintain cooperation[26,44], has been recently re-visited and eventually ruled out by thorough comparative genetic analysis[45]. While likely rare it is possible that in nature, revertants could occur by other means than conjugation like generalized transduction of wild-type cooperative alleles in the place of mutant alleles. Our experimental work nevertheless strengthens the notion that, if HGT may allow for the emergence of cooperators, it is unlikely that it ensures the long-term stability of cooperative traits. The maintenance of cooperative virulence rather depends on additional factors, including managing the cost of virulence expression via tight regulation[16,46], and the transmission dynamics of the pathogen (i.e., timing and population bottleneck size) between hosts.

What about a potential role of tissue invasion and the subsequent formation of intra-cellular *Salmonella* populations in these dynamics? In this study, we use TTSS-2 mutants to prevent deadly systemic spread of S.Tm in C57BL/6J Nramp knock-out mice[47]. The TTSS-2 mutation limits the intra-cellular population size of S.Tm[32,47]. In theory, this might diminish the frequency at which intra-cellular bacteria travel back from the tissues into the intestinal lumen[48]. However, we have previously demonstrated that, in the streptomycin pretreated mouse model, the intracellular reservoir of virulent clones is only relevant for stabilizing cooperation when the intestinal lumen is empty, for instance after treating mice with an antibiotic. This allows luminal growth of re-seeding bacteria from the tissues[23]. In the absence of antibiotic-mediated depletion of

the gut-luminal S.Tm population, cheaters outcompete cooperators as observed in this study. Moreover, once cheaters are fixed in the luminal population, reseeding events of cells carrying the cooperative allele on MGE[31] or on the chromosome[23] will be too rare to restore virulence. Based on these considerations, it is reasonable to think that the choice of a TTSS-2 defective background strain does not fundamentally affect the general conclusion about the insufficiency of HGT to restore virulence in the long run. Higher fitness cost in our experimental system led to greater instability of cooperative virulence (Fig. 1), in line with the fact that a complex fine-tuned regulation is essential to ensure virulence stability in S.Tm[11] and other "cooperative" pathogens[49–51]. Accordingly, *hilD* is integrated into the chromosome of S.Tm within SPI-1, which, compared to carriage on multi-copy plasmids, could favor the evolution of a regulation deeply entangled with the general physiology of the pathogen[16,46,52].

Long-term intestinal colonization was detrimental for cooperative virulence as fast growing cheaters accumulate over generations within-host (Figs. 1–4)[11,23]. Theoretical work predicts that transmission likely plays a role in the epidemiological success of virulence[4,24,53]. In our case, early transmission ensures that enough cooperators can trigger the disease in the next host (Fig. 2). Moreover, beside timing, population bottlenecks at transmission influence this process[10]. S.Tm is a case where the benefit of cooperative virulence directly fuels transmission[37]. Cheaters alone cannot free-ride off of the inflammation normally triggered by cooperators that increases the carrying capacity in the gut and transmission of the pathogen. Hence, narrow population bottlenecks at transmission favor cooperation, as described in other experimental systems[54,55], and could apply for other enteric pathogens such as *Vibrio cholerae* (encoding the cholera toxin on a phage)[56], *Shigella* spp. (containing phage- and plasmid-encoded secreted virulence factors)[57], or *Yersinia* spp.

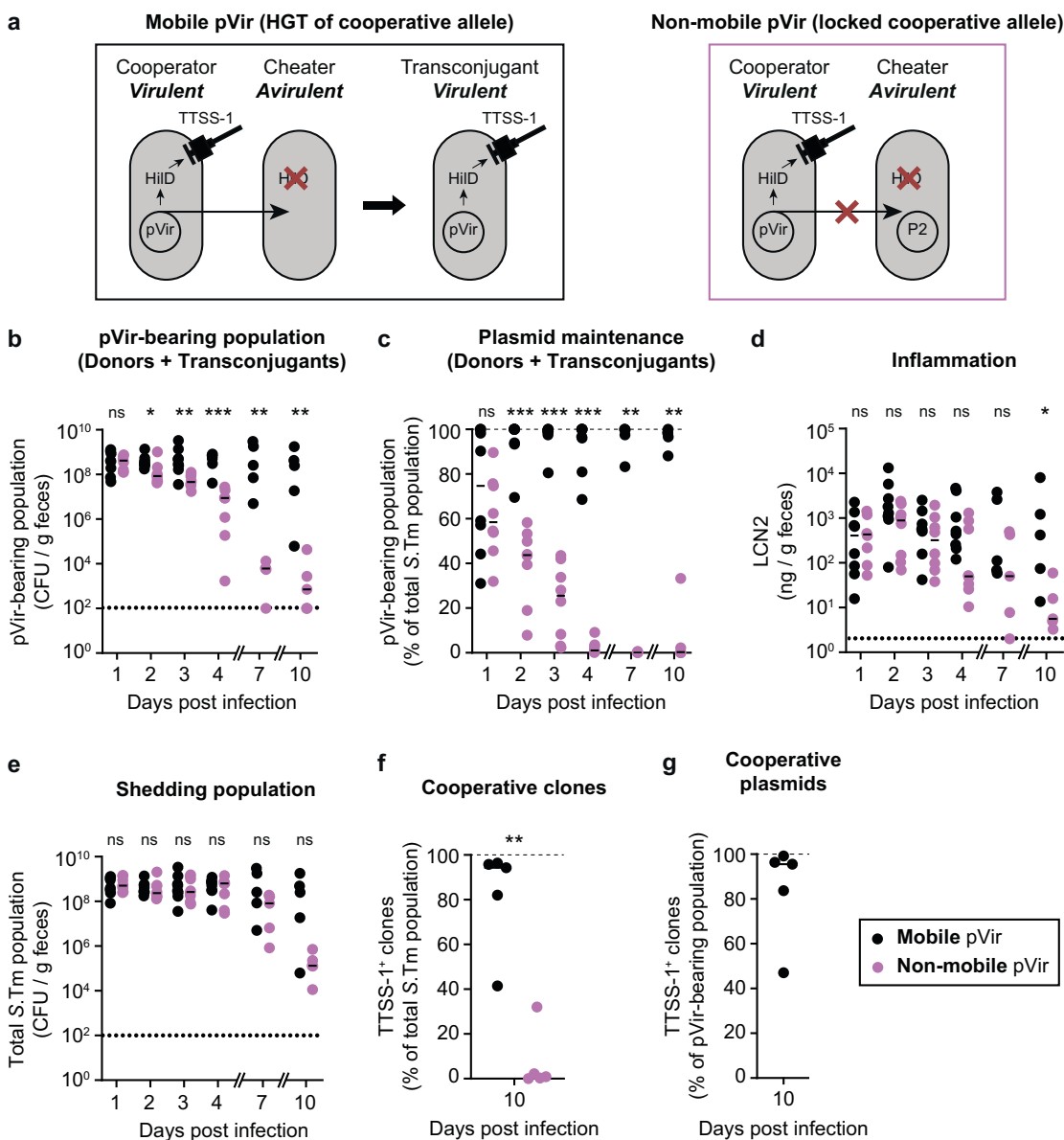

**Fig. 4 HGT can increase the duration of, but does not stabilize, cooperative virulence. a** Experimental system to determine the role of HGT in the stabilization of cooperative virulence. In both the mobile pVir and non-mobile pVir scenarios, the cooperator contains pVir$^{Low}$ and a functional TTSS-1, making it virulent. In the mobile scenario, pVir can be transferred to cheaters. In the non-mobile scenario, transfer of pVir is blocked because of an incompatible plasmid, P2, in the cheater strain. **b–g** Ampicillin pretreated mice were orally infected with ~$10^2$ CFU of cheater (14028 Δ*hilD* Δ*ssaV*; Kan$^R$, Amp$^R$) immediately followed by ~$10^2$ CFU of pVir$^{Low}$ donor (14028 Δ*hilD* Δ*ssaV* pVir; Cm$^R$ encoded on pVir, Amp$^R$). The donor contained a functional *invG* allele, making it virulent (*ssaV* is deleted in both strains). Mice (*n* = 8 until day 4; *n* = 5 until day 10 for both groups) were given either a cheater with no plasmid (i.e., same strain used in Fig. 1; mobile pVir; black) or a cheater with P2 (incompatible with pVir; non-mobile pVir; pink). Dotted lines indicate detection limits. Medians are indicated by lines. Two-tailed Mann–Whitney *U* tests (*p* > 0.05 (ns), **p* < 0.05, ***p* < 0.01, ****p* < 0.001) are used to compare the mobile pVir and non-mobile pVir scenarios on each day. Donor, recipient, and transconjugant populations are presented in Fig. S6. Source data are provided as a Source Data file. **b** The pVir-bearing population was determined by selective plating on Cm-supplemented MacConkey agar. **c** the pVir-bearing population is reported as a percentage of the total population. Dashed line indicates 100% plasmid spread. **d** Inflammation was measured by a Lipocalin-2 ELISA on fecal samples. **e** Total *S*.Tm populations determined by the sum of Cm- and Kan-supplemented MacConkey agar. **f**, **g** Cm-supplemented MacConkey agar plates containing colonies from feces collected on day 10 post infection were analyzed for SipC expression as a proxy for TTSS-1 expression using a colony western blot and represented as the percentage of colonies that produced SipC are reported out of the total *S*.Tm population (**f**) or out of the pVir-containing population for the mobile pVir scenario (**g**).

(encoding the Yop virulon on a plasmid)[58], which all use secreted virulence factors to survive in the host and/or directly increase shedding. In the case of *S*.Tm, population fragmentation occurs when a subset of the population is released from a host through fecal pellets, and bottlenecks are likely to occur in harsh external environment or during colonization of a recipient host in which colonization resistance is mediated by the protective gut microbiota. Environmental factors such as diet perturbations[28,59] or exposure to antibiotics[35], that affect the composition of the microbiota and reduce colonization resistance, can widen transmission bottlenecks and should have profound implications on the evolution of virulence in *S*.Tm. Further work is needed to

**Table 2 Plasmids used in this study.**

| Plasmid name | Relevant genotype | Resistance | Ref. |
|---|---|---|---|
| pM975 | *bla*; used to confer ampicillin resistance | Amp | 32 |
| pCP20 | FLP recombinase | Amp, Cm | 63 |
| pKD46 | Arabinose-incudible *λ red* system | Amp | 63 |
| pM972 | $P_{sicA}$-*gfp*; reporter for TTSS-1 | Amp | 19 |
| pKD3 | *cat* | Cm | 63 |
| pKD3-*hilD-cat* Low | *hilD* with 648 bp of upstream regulatory region, *cat* | Cm | This work |
| pKD3-*hilD-cat* High | *hilD* with 279 bp of upstream regulatory region, *cat* | Cm | This work |
| P2 | Wild-type | None | 14 |
| P2*cat* | *cat* | Cm | 14 |
| pVir^Low | *hilD* with 648 bp of upstream regulatory region, *cat* | Cm | This work |
| pVir^High | *hilD* with 279 bp of upstream regulatory region, *cat* | Cm | This work |

fathom this key aspect of the host/pathogen/microbiota interaction.

The current antibiotic crisis[60] highlight the critical importance of understanding the role of bacterial sociality in the evolution of virulence, as this aids the discovery of antibiotic-free treatments to manage bacterial infections. It has inspired "anti-virulence compounds" that target extracellular virulence factors or their expression as a therapeutic avenue for minimizing resistance (reviewed in[61]. Alternatively, exploiting cheating behavior to destabilize cooperation has been suggested as a possible therapeutic strategy (coined "Hamiltonian medicine")[62]. Previous work on S.Tm[23] demonstrated that administering *hilD* mutants has the potential to prevent inflammation-mediated *Salmonella* blooms and transmission. The present study suggests that, in the absence of direct positive selective pressure within-host, HGT of functional *hilD* copies is rather unlikely to restore virulence in *hilD* mutants. Indeed, preventing inflammation and S.Tm bloom also prevents HGT in the gut[4,14]. Besides, providing that such cooperative alleles would naturally exist on MGEs, the cost of suboptimal virulence expression levels makes them prone to inactivation, without impairing their ability to spread in the recipient population once inactivated. Furthermore, strategies that slow down HGT, such as vaccination[29–31], could be used in combination with avirulent competitors to further reduce the horizontal spread of cooperative alleles, thus ensuring evolutionary robustness of cheater-based biocontrol approaches.

## Methods

**Strains, plasmids, and primers used in this study.** All the strains and plasmids used in this work are summarized in Tables 1 and 2. Bacteria were grown in lysogeny broth (LB) media containing the appropriate antibiotics (50 μg/ml streptomycin (AppliChem); 15 μg/ml chloramphenicol (AppliChem); 50 μg/ml kanamycin (AppliChem); 100 μg/ml ampicillin (AppliChem)) at 37 °C or 30 °C if containing pKD46 or pCP20. Gene deletion mutants were performed using the *λ red* system[63]. Desired genetic constructs were transferred into the appropriate background strain using P22 HT105/1 *int-201* phage transduction[64]. Antibiotic resistance cassettes were removed using the heat-inducible FLP recombinase encoded on pCP20, if desired[63]. Expression vectors (e.g. pM975 and pM972) were transformed into the desired strain using electroporation.

To create pVir donor strains, *hilD* was amplified with PCR using high-fidelity Phusion polymerase (ThermoFisher Scientific) from the chromosome of SL1344 with either 648 bp (low cost) or 279 bp (high cost) of regulatory region (Fig. S1) and cloned into pKD3 upstream of the chloramphenicol resistance cassette using Gibson Assembly (NEB). Primers to amplify *hilD* contained ~40 bp homology to the sites flanking a *Nde*I site in pKD3. pKD3 was digested with *Nde*I, purified, and mixed with the PCR amplicon in Gibson Assembly Master Mix (NEB; protocol as described by the manufacturer). The products were transformed into *E. coli* CC118 *λpir*, and colonies were verified to contain the desired plasmid through PCR and Sanger sequencing. The resulting *hilD-cat* construct was then amplified from cloned plasmid with Phusion PCR using primers with homology to the target site in P2 (upstream of the colicin Ib locus; *cib*) and introduced into SB300 Δ*hilD* using *λ red*[63]. Positive clones were determined by PCR, leading to pVir^Low and pVir^High. Lastly, the pVir plasmids were conjugated in vitro into the desired strain by mixing the 10^5 CFU from an overnight culture of the donor strain with the desired recipient, allowing conjugation overnight at 37 °C on a rotating wheel, and plating

the cells on MacConkey agar to select for transconjugants. For in vivo experiments, pVir plasmids were conjugated into 14028 S Δ*hilD* Δ*invG* Δ*ssaV* pM975 or 14028 S Δ*hilD* Δ*ssaV* pM975. For TTSS-1 expression analysis, pVir plasmids were conjugated into 14028 S Δ*hilD ssaV::aphT* pM972. All primers used for strain or plasmid construction and verification are listed in Table 3.

**In vitro growth and TTSS-1 expression.** Subcultures were grown in LB with appropriate antibiotics for 6 h and subsequently diluted 200 times in 200 μl of media distributed in 96-well black side microplates (Costar). The lid-closed microplates were incubated at 37 °C with fast and continuous shaking in a microplate reader (Synergy H4, BioTek Instruments). Optical density at 600 nm and GFP fluorescence (491 nm excitation; 512 nm emission) were measured every 10 min for 14 h. OD and fluorescence values were corrected for the baseline value measured for sterile broth.

**Infection experiments.** All mouse experiment protocols are derived from the streptomycin pretreated mouse model described in Barthel et al.[35]. We used ampicillin rather than streptomycin since S.Tm 14028S is not naturally resistant to streptomycin. Ampicillin resistance is conferred by pM975 contained in all strains used in vivo. All experiments were performed in 8–12-week-old specified opportunistic pathogen-free (SOPF) C57BL/6J mice, which were given 20 mg of ampicillin by oral gavage to allow robust colonization of S.Tm. This ampicillin pretreatment model has been used previously to measure HGT in the gut[29–31]. All infection experiments were approved by the responsible authorities (Tierversuchskommission, Kantonales Veterinäramt Zürich, licenses 193/2016 and 158/2019). Sample size was not predetermined and mice were randomly assigned to treatment group.

*Single infections of donors or recipients.* Overnight cultures grown at 37 °C in LB with the appropriate antibiotics were diluted 1:20 and subcultured for 4 h in LB without antibiotics. Cells were centrifuged and resuspended in PBS before being diluted. Ampicillin pretreated mice were orally gavaged with ~5 × 10^7 CFU. Fecal samples were collected daily, homogenized in PBS with a steel ball at 25 Hz for 1 min, and bacterial populations were enumerated on selective MacConkey agar. Lipocalin-2 ELISA (R&D Systems kit; protocol according to manufacturer) was performed on feces to determine the inflammatory state of the gut. At day 4 post infection, mice were euthanized.

*Plasmid transfer experiments.* Donor and recipient strains (14028S derivatives; *ssaV* mutants) were grown overnight in LB with the appropriate antibiotics at 37 °C and subsequently diluted 1:20 and subcultured for 4 h in LB without antibiotics, washed in PBS, and diluted. Ampicillin pretreated mice were orally gavaged sequentially with ~10^2 CFU of donors followed by ~10^4 CFU of recipients. Feces were collected when needed, homogenized in PBS, diluted, and bacterial populations were enumerated on MacConkey agar containing the appropriate antibiotics (donors = Cm; recipients = Kan; transconjugants, i.e., a recipient (kanamycin resistant, kan) carrying the plasmid and with it an extra-resistance to chloramphenicol (Cm) (selection Cm+Kan); total population = populations of donors + recipients). Plating allows analyzing up to 200 clones. Therefore, when 99% of recipients are transconjugants, we count 2 KanR/CmS clones for 198 KanR/CmR clones, i.e., more transconjugants than recipients without plasmid. The detection limit was a 2-log difference between transconjugants and the remaining fraction of recipient cells. Replica plating was used if the CFUs on the Cm+Kan plates approached those on the Cm or Kan plates to determine an exact ratio of plasmid transfer, and the donor population size. At day 10 post infection, mice were euthanized. Lipocalin-2 ELISA was performed on feces to determine the inflammatory state of the gut. When needed, transconjugants on the Cm+Kan plates were kept at 4 °C until analysis by colony blot.

**Table 3 Primers used in this study.**

| Primer name | Sequence (5′ to 3′) | Purpose | Ref. |
|---|---|---|---|
| HilD-31-F | GGAACACTTAACGGCTGACATGGGAATTAGCCATGGTCCATACAGGATAAGCAATTCACCG | Gibson Assembly of *hilD* into pKD3 (pVir$^{Low}$) | This work |
| HilD-1-F | GGAACACTTAACGGCTGACATGGGAATTAGCCATGGTCCATAGCAGATTACCGCACAGGA | Gibson Assembly of *hilD* into pKD3 (pVir$^{High}$) | This work |
| HilD-2-R | AGAATAGGAACTTCGGAATAGGAACT AAGGAGGATATTCATAGTGTTAATGC GCAGTCTGA | Gibson Assembly of *hilD* into pKD3 (both pVir$^{Low}$ and pVir$^{High}$) | This work |
| HilD-3Po-F | AGGAACTTCGGAATAGGAAC | Verification of *hilD* in pKD3 | This work |
| HilD-4Po-R | AACACTTAACGGCTGACATG | Verification of *hilD* in pKD3 | This work |
| HilD-32-F | GCATGATAATAATAATCAATAACAATAAGCTGTGTCACGTTTACATCATCAGGATAAGCAATTCACCG | λ-red for *hilD-cat* in P2 downstream of *cib* (pVir$^{Low}$) | This work |
| HilD-29-F | GCATGATAATAATAATCAATAACAATAAGCTGTGTCACGTTTACATCATGCAGATTACCGCACAGGA | λ-red for *hilD-cat* in P2 downstream of *cib* (pVir$^{High}$) | This work |
| HilD-30-R | AAGGGTAATGGCGGAAGCCGGATACCCAGCCGCCAGAGAATGTGTAGGCTGGAGCTGCTTC | λ-red for *hilD-cat* in P2 downstream of *cib* (both pVir$^{Low}$ and pVir$^{High}$) | This work |
| insert_ p2_up | GTA CCG GTG CGT GAT AAC | Verification of *hilD-cat* insert in P2 to create pVir (both pVir$^{Low}$ and pVir$^{High}$) | [31] |
| insert_ p2_dw | CAA CAG CGT GAC CTG CC | Verification of *hilD-cat* insert in P2 to create pVir (both pVir$^{Low}$ and pVir$^{High}$) | [31] |
| ver_*hilD*_up2 | TCTCGATAGCAGCAGATTAC | Verification of Δ*hilD* in the chromosome | [11] |
| ver_*hilD*_dw2 | CAGTATAAGCTGTCTTCCG | Verification of Δ*hilD* in the chromosome | [11] |
| *ssaV*-137F | GCAGCGTTCCAGGGTATTCC | Verification of Δ*ssaV* in the chromosome | This work |
| *ssaV*+155R | CAGCAAGTTCTTCTCCAGGC | Verification of Δ*ssaV* in | This work |

**Table 3 (continued)**

| Primer name | Sequence (5′ to 3′) | Purpose | Ref. |
|---|---|---|---|
| | | the chromosome | |
| *invG*-134F | GAAGGCCACGAGAACATCAC | Verification of Δ*invG* in the chromosome | This work |
| *invG* + 112R | GCGGCCTGTTGTATTTCCGC | Verification of Δ*invG* in the chromosome | This work |

*Competitions involving mobile versus non-mobile pVir.* Cooperator (donor) and cheater (recipient) strains (14028S derivatives; *ssaV* mutants) were grown overnight in LB with the appropriate antibiotics at 37 °C and subsequently diluted 1:20 for a 4 h subculture in LB without antibiotics. Cells were resuspended in PBS and diluted. Ampicillin pretreated mice were sequentially orally gavaged with ~$10^2$ CFU of cooperators (pVir$^{Low}$) immediately followed by ~$10^2$ CFU of cheaters. Feces were collected when needed, homogenized, diluted, and bacterial populations were enumerated on MacConkey agar containing the appropriate antibiotics as for the plasmid transfer experiments. At day 4 or 10 post infection, mice were euthanized. Lipocalin-2 ELISA was performed on feces to determine the inflammatory state of the gut. The colonies on the Cm plates from day 10 fecal plates were kept at 4 °C until analysis by colony blot.

*Transmission experiments.* Feces from mice given donor and recipient strains were collected on day 2 and day 10 p.i., resuspended in PBS, briefly centrifuged, and 100 μl of the suspension was given to ampicillin pretreated mice. These experiments occurred in parallel to the plasmid transfer experiments to ensure fresh fecal populations were transmitted into new mice. Bacterial populations and the state of inflammation were measured as for the plasmid transfer experiments. Mice were euthanized at day 4 post transmission.

*Evolved transconjugant infections.* Single clones from plasmid transfer experiments were isolated on day 7 or day 10, and stored in 20% LB + glycerol at −80 °C. Isolates were grown in LB containing the appropriate antibiotics (Cm, Kan, Amp) overnight at 37 °C and subsequently diluted 1:20 and subcultured for 4 hours in LB without antibiotics. Of note, loss of pM975 was observed for some clones (based on loss of ampicillin resistance), and could therefore not be used for infection. Subcultured cells were centrifuged, resuspended in PBS, and ~$5 \times 10^7$ CFU were given to ampicillin pretreated mice by oral gavage. The shedding population was enumerated on MacConkey supplemented with chloramphenicol after suspension in PBS followed by dilution. On day 4 post infection, mice were euthanized and fecal samples were additionally enumerated on MacConkey supplemented with kanamycin, to ensure that plasmid loss did not contribute to the detected shedding population. The kanamycin-resistant colonies (Kan$^R$ is encoded on the chromosome) were replica plated onto MacConkey supplemented with chloramphenicol to confirm that no pVir plasmid loss occurred. The MacConkey chloramphenicol plates were stored at 4 °C until analysis by colony blot. LCN2 ELISA was used to determine the inflammatory state of the mice over time.

**Colony blots**. To assess TTSS-1 expression at the clonal level (to determine the proportion of cooperators), a colony Western blot was performed. SipC was used as a proxy for TTSS-1 expression, since SipC is regulated by HilD. We have previously established this protocol to assess heterogeneously expressed phenotypes such as TTSS-1 in *S*.Tm[11,27], since single-cell approaches would not differentiate cheaters from the phenotypically OFF subpopulation[27]. For a detailed protocol and an overview of applications, see[27]. Briefly, colonies on MacConkey agar were replica transferred to nitrocellulose membranes and placed face-up on LB agar without antibiotics and allowed to grow overnight. The original MacConkey plates are also allowed to re-grow and then stored at 4 °C. Colonies were lysed and cellular material was hybridized to the membrane by passing the membranes over a series of Whatman filter papers soaked with buffers: 10 minutes on 10% SDS, 10 minutes on denaturation solution (0.5 M NaOH, 1.5 M NaCl), twice for 5 minutes on neutralization solution (1.5 M NaCl, 0.5 M Tris-HCl, pH 7.4), and 15 minutes on 2× SSC (3 M NaCl, 0.3 M sodium citrate, pH 7). Membranes were washed twice with TBS (10 mM Tris-HCl, 150 mM NaCl, pH 7.4) and excess cellular debris was gently removed by scraping the surface with a folded Whatman paper. Membranes were blocked with TBS containing 3% BSA for 1 h at room temperature and then incubated with 5 ml of TBS with 3% BSA containing a 1:4000 dilution of anti-SipC rabbit antibody provided by Virotech Diagnostics GmbH (reference number: VT110712) overnight in a moist chamber at 4 °C on a rocking platform. Washing once with TBS-T (20 mM Tris-HCl, 500 mM NaCl, 0.05% Tween 20, 0.2% Triton X-100, pH 7.5) and twice with TBS removed non-specific binding. Secondary

antibodies (1:2500 dilution of goat anti-rabbit IgG conjugated to HRP; Sigma; catalog number A0545-1ML) were then added to membrane in TBS with 3% BSA and incubated at room temperature on a rocking platform for 2-4 hours. Three more washing steps with TBS were performed before resolving the staining with 5 ml of substrate per membrane: a 30 mg tablet of 4-chloro-1-naphthol (Sigma) dissolved in 10 ml of methanol, mixed with $H_2O_2$ (0.06% w/v) in 50 ml of TBS. The reaction is stopped with water after the desired intensity is observed.

Clones of interest can be identified by changes in SipC abundance. Desired isolates were matched to the original MacConkey plate and inoculated in LB containing chloramphenicol and kanamycin. Isolates were then stored in 20% LB + glycerol at −80 °C until whole-genome bacterial sequencing was performed, or evolved clones were used for infection.

**Whole-genome bacterial sequencing**. Strains stored in 20% LB + glycerol at −80 °C were inoculated in LB with the appropriate antibiotics. Genomic DNA was extracted from 1 ml of overnight culture using a QIAamp DNA Mini Kit (Qiagen). Illumina MiSeq sequencing operated by the Functional Genomics Centre Zurich and Novogene (Cambridge) was performed to generate 150 bp paired end reads with at least 50× coverage across the genome. Bioinformatic analysis was performed using CLC Genomics Workbench 11.0. Reads were mapped to the 14028S chromosome reference (NCBI accession NC_016856.1 [https://www.ncbi.nlm.nih.gov/nuccore/NC_016856.1/]) and the pVir plasmids (the SL1344 P2 plasmid (NCBI accession NC_017718.1 [https://www.ncbi.nlm.nih.gov/nuccore/NC_017718.1/]) was modified by inserting the cloned *hilD-cat* regions to create pVir$^{Low}$ and pVir$^{High}$ reference sequences). Basic variant detection was performed to detect variants that occurred in a minimum of 70% of reads. Variants were excluded if they occurred in non-specific regions determined by read mapping in CLC (e.g. where reads could map equally well to another location in the genome). Small insertions or deletions (Indels) were also detected using software in CLC Genomics Workbench 12.0.2. This is summarized in Tables S1–4.

**Statistical analysis**. Data were collected using Microsoft Excel 2016 (16.0.5266.1000). Statistical tests on experimental data were performed using GraphPad Prism 8.0.2 for Windows.

**Reporting summary**. Further information on research design is available in the Nature Research Reporting Summary linked to this article.

## Data availability

All source data to generate plots in the manuscript and *p*-values for statistical comparisons are available in the source data Excel sheet. Citation of the current paper should accompany any further publication based on these data. Figures 1–4 and S1–S6 are associated with raw data. All Illumina sequencing data are publicly available from the NCBI repository (BioProject accession number PRJNA817059). Source data are provided with this paper.

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

## Acknowledgements

We would like to thank members of the Hardt and Diard labs for valuable discussions, as well as the staff at RCHCI and EPIC animal facilities. We thank Stuart West, Kevin Foster and Ashleigh Griffin for helpful feedback and comments on this manuscript. We would also like to acknowledge the staff at Functional Genomics Centre Zurich and Novogene for whole-genome bacterial sequencing. We acknowledge grant funding from the Swiss National Science Foundation (NRP 72 407240_167121, 310030B_173338, and 310030_192567), the Gebert Rüf Foundation (GRS-060/18) and the Monique Dornon-ville de la Cour Foundation to WDH. MD is funded by an SNF professorship grant (PP00PP_176954) and a BRCCH multi-investigator grant, J.S.H. is funded by NRP72 grant 407240-167121 from the Swiss National Science Foundation, and EB received a Boehringer Ingelheim Fonds PhD fellowship.

## Author contributions

E.B., W.D.H., and M.D. conceived the project and designed the experiments. E.B., E.G., Y.S., and A.R. carried out the experiments and analyzed data. E.B., W.D.H., and M.D. wrote the manuscript. E.G. and J.S.H. provided valuable input on experimental design, theoretical background, and the manuscript. All authors read, commented on, and approved this manuscript.

## Competing interests

The authors declare no competing interests.
