## [Peer Review File · Nature Communications]

Impact of horizontal gene transfer on emergence and stability of cooperative virulence in *Salmonella Typhimurium*Reviewers' Comments:

Reviewer #1:

Remarks to the Author:

This is an extremely thorough manuscript investigating the importance of Horizontal Gene Transfer (HGT) for the maintenance of cooperative virulence in the extensively studied *Salmonella* Typhimurium-mouse model system. The work sets out to test theories about the importance of HGT in the maintenance of cooperation. Despite extensive conjugation rates, and consideration of both within and between-host dynamics, the main conclusion is that HGT is relatively unimportant. I cannot fault the experiments, writing or interpretation.

It could be argued that these results are not especially novel, as they essentially recapitulate the authors previous work where the *hilD* is chromosomally encoded. I think it is important to be really explicit about this. However, showing that this cooperative transmission is relatively unimportant in a natural setting is an important contribution, as previous work (cited) is correlational or in vitro. Furthermore, the reason why HGT does not maintain cooperation is because of the invasion of "cheating" plasmids, as predicted by theory in relatively well-mixed environments (<https://www.ncbi.nlm.nih.gov/pmc/articles/PMC3652439/#RSPB20130400C39>). I think the authors could again highlight this more clearly.

Reviewer #2:

Remarks to the Author:

In the work by Bakkeren et al., the authors tracked the emergence of cooperative virulence traits in the *Salmonella* Typhimurium gut population. This pathogen employs the type III secretion system (T3SS), encoded for by SPI-1, to invade the host cells prior to establishing an intracellular niche using a second T3SS. The fitness cost of expressing T3SS-1 results in the emergence of lineages that carry mutations inactivating the master transcriptional regulator, *HilD*. While more fit than their ancestor, the evolved lineages are defective in T3SS expression, which curtails their ability to trigger inflammation and cause disease. Given that inflammation is required for the expansion of *Salmonella* in the gut and for host transmission, the *hilD*- lineages co-exist as cheaters with the *hilD*+ clones, suggesting that the bacterial population is under negative frequency-dependent selection in vivo. Using conjugative plasmids, the authors showed that cheaters can acquire *hilD* from isogenic donor strains in the gut, which restored the expression of T3SS and the associated pro-inflammatory traits. This work investigates the role of HGT in the emergence of virulence among gut bacteria. Additionally, the study provides some insights into the evolutionary forces driving population heterogeneity in *Salmonella*, potentially uncovering new targets for anti-infective agents and/or biocontrol. Overall, the study could be of interest to the infection biology readership. However, I have the following major concerns:

1-The authors employed a modified conjugative plasmid to demonstrate the transmission of *hilD* to cheaters. While I understand the need for a contrived system to track bacterial evolution in vivo, the model excludes other forms of HGT that are known to influence the population dynamics of *Salmonella*. The authors showed before that inflammation activates phage transduction during *Salmonella* infection (Diard et al., 2017). Thus, why did the authors focus solely on conjugation as an evolutionary force? If *hilD* were to be transmitted via the *SopE ϕ* prophage, would the results (cooperator emergence, fitness cost...etc) have varied? It is possible that *hilD* and another plasmid-encoded element have a cumulative negative effect on fitness, contributing to the reversion of cooperative virulence in the transconjugants. It is also possible that transduction will occur at a different rate than conjugation, which can influence the stability of the heterogeneous population. The authors should share their rationale with regards to favoring one form of HGT over another.

2-The sequencing of clinical isolates in previous studies showed that *hilD*- cheaters emerge frequently

in the Salmonella in vivo-population. In this regard, the authors predicted that HGT can facilitate the reversal of these cheaters to virulent clones. Were there any revertants detected among the clinical isolates? I understand that the authors employed Salmonella as a tractable model to make inferences about other systems. However, it will be interesting if there is precedence for virulence restoration in cheaters among the natural isolates of Salmonella.

3- The authors used SPI-2 T3SS- Salmonella strains to infect C57BL/6 mice (Nramp-). This is to reduce the inflammation and avoid the rapid death of the host. Why is this a better strategy than infecting 129 SvEv Nramp+ mice with SPI-2 T3SS+ Salmonella? The Nramp+ vacuoles in the latter mouse strain should limit Salmonella expansion, supporting infection for longer time periods than C57BL/6 without the need to inactivate SPI-2 T3SS. The latter is required by Salmonella strains to establish their intravacuolar habitat. This allows the establishment of intracellular reservoirs from which a subset of the bacteria will escape and hyperreplicate in the cytoplasm of epithelial cells (Klein et al., 2017). Given that SPI-1 T3SS was shown to be important for the expansion of the cytosolic Salmonella population, this phase of the infection may select for the hiiD+ co-operative clones. In this regard, the model employed by the authors is potentially lacking some relevant selective forces, which can impact the findings. This needs to be addressed.

4-FigS1: The authors showed a correlation between T3SS-1 expression and growth defect. They attributed the fitness cost associated with pVir-high to the overexpression of T3SS-1, mediated by HiiD. What about the other HiiD-regulated genes, like flagellar motility genes? How much do they contribute to the fitness defect? The authors did not acknowledge this possibility. If this was addressed in previous studies, the authors should mention that.

5- The authors infected mice with clonal Salmonella populations to mimic the effects of genetic drift during infection transmission. It will also be interesting to investigate the population dynamics during natural transmission (e.g. co-housing abx-treated animals with the infected ones). This will be a better proxy to what occurs in nature.

Minor comments:

Fig S3A: Why are there more transconjugants than recipients at some time points (e.g. day 2)? The authors mentioned that transconjugant enumeration was done after replica plating. Please clarify.

Fig 1F: This figure doesn't add new information, so please remove it or move to the supplementary data. The trends are clear in the figures preceding this one.

Fig 3D-F: The information in these figures can be inferred from the other data. This sort of redundancy is distracting to the reader. Also, the positive correlations presented in 3E and 3F are weak despite being significantly different from zero.

Lines 61: The authors used "cooperative virulence" to refer to the co-expression of hiiD in trans and T3SS-1 in cis. However, the reader can confuse this with lineage co-existence (e.g. the co-existence of the virulent hiiD+ clones and the hiiD- cheaters). This needs to be clearly explained in the manuscript to make it easy for the reader to follow the narrative.

Line 125: How many evolved isolates were sequenced?

Line 128: Please indicate that 'MGE' stands for 'mobile genetic elements'

Line 128: The authors described the slower loss of T3SS-1+ clones in the pVir-low population compared to pVir-high counterpart as 'striking'. Isn't that expected given that the fitness cost of expressing HiiD is higher in the latter?

Line 136: Change "with" to "by"

Lines 175-176: "we concluded that the proportion of cooperators do contribute to disease transmission." Change this sentence to ".....contribute to disease development post transmission" because transmission in this experiment was artificial. Thus, the emphasis should be on how the proportion of cooperators affects inflammation development. Any claims regarding the efficiency of pathogen transmission between hosts are speculative.

Line 177: What is the "sufficient" proportion of cooperative clones required in the population to trigger inflammation? This requires testing different proportions of the cooperative lineages.

Lines 278-280: The authors mentioned anti-virulence compounds, but then suggested the administration of cheaters to destabilize the Salmonella population. The latter is a form of biocontrol; hence, this part is confusing. A discussion of compounds that can potentially accelerate the emergence of cheater/avirulent clones will be more appropriate.

Line 455: Please indicate which mouse strain was used, C75BL/6 J or N.

NCOMMS-21-28617A

Point-by-point response to the reviewers' comments

Reviewer #1 (Remarks to the Author):

This is an extremely thorough manuscript investigating the importance of Horizontal Gene Transfer (HGT) for the maintenance of cooperative virulence in the extensively studied *Salmonella* Typhimurium-mouse model system. The work sets out to test theories about the importance of HGT in the maintenance of cooperation. Despite extensive conjugation rates, and consideration of both within and between-host dynamics, the main conclusion is that HGT is relatively unimportant. I cannot fault the experiments, writing or interpretation.

It could be argued that these results are not especially novel, as they essentially recapitulate the authors previous work where the *hilD* is chromosomally encoded. I think it is important to be really explicit about this. However, showing that this cooperative transmission is relatively unimportant in a natural setting is an important contribution, as previous work (cited) is correlational or *in vitro*. Furthermore, the reason why HGT does not maintain cooperation is because of the invasion of "cheating" plasmids, as predicted by theory in relatively well-mixed environments (<https://www.ncbi.nlm.nih.gov/pmc/articles/PMC3652439/#RSPB20130400C39>). I think the authors could again highlight this more clearly.

Response: We would like to thank reviewer 1 for the positive assessment of our work. We agree with the reviewer that the evolution of cheaters by mutating the mobile cooperative allele is an expected outcome in line with theoretical works and the emergence of cheaters by chromosomal mutations previously observed during infection. We also agree that the strength of our study is to demonstrate experimentally that a mobile cooperative allele does not stabilize cooperative virulence in the long run *in vivo*.

As suggested by reviewer 1, we have modified the introduction and discussion to better present this work in the light of previous theoretical and experimental contributions (lines 63 and 252-258).

Reviewer #2 (Remarks to the Author):

In the work by Bakkeren et al., the authors tracked the emergence of cooperative virulence traits in the *Salmonella* Typhimurium gut population. This pathogen employs the type III secretion system (T3SS), encoded for by SPI-1, to invade the host cells prior to establishing an intracellular niche using a second T3SS. The fitness cost of expressing T3SS-1 results in the emergence of lineages that carry mutations inactivating the master transcriptional regulator, *HilD*. While more fit than their ancestor, the evolved lineages are defective in T3SS expression, which curtails their ability to trigger inflammation and cause disease. Given that inflammation is required for the expansion of *Salmonella* in the gut and for host transmission, the *hilD*- lineages co-exist as cheaters with the *hilD*+ clones, suggesting that the bacterial population is under negative frequency-dependent selection *in vivo*. Using conjugative plasmids, the authors showed that cheaters can acquire *hilD* from isogenic donor strains in the gut, which restored the expression of T3SS and the associated pro-inflammatory traits.

This work investigates the role of HGT in the emergence of virulence among gut bacteria. Additionally, the study provides some insights into the evolutionary forces driving population heterogeneity in *Salmonella*, potentially uncovering new targets for anti-infective agents and/or biocontrol. Overall, the study could be of interest to the infection biology readership.

Response: We would like to thank reviewer 2 for helpful comments and suggestions.

However, I have the following major concerns:

1-The authors employed a modified conjugative plasmid to demonstrate the transmission of *hilD* to cheaters. While I understand the need for a contrived system to track bacterial evolution *in vivo*, the model excludes other forms of HGT that are known to influence the population dynamics of *Salmonella*. The authors showed before that inflammation activates phage transduction during *Salmonella* infection (Diard et al., 2017). Thus, why did the authors focus solely on conjugation as an evolutionary force? If *hilD* were to be transmitted via the SopE ϕ prophage, would the results (cooperator emergence, fitness cost...etc) have varied?

It is possible that *hilD* and another plasmid-encoded element have a cumulative negative effect on fitness, contributing to the reversion of cooperative virulence in the transconjugants. It is also possible that transduction will occur at a different rate than conjugation, which can influence the stability of the heterogeneous population. The authors should share their rationale with regards to favoring one form of HGT over another.

Response: Reviewer 2 suggests that phage-mediated gene transfer might result in a different outcome than conjugation. This is a fair point and, in fact, testing SopE ϕ as a vector for the cooperative allele *hilD* was our first attempt to evaluate the impact of HGT on the evolution of cooperative virulence. The data obtained with SopE ϕ -*hilD* (ϕ vir) (robust but too preliminary for publication) are presented in the figure below. The transfer rate of ϕ vir was too slow and unreproducible *in vivo* to be exploited. Our previous work suggests that this low efficiency is attributable, at least in part, to the need for gut inflammation to initiate the lytic cycle of SopE ϕ (Diard et al., Science 2017). This initial inflammation is lacking in our current experimental setup, that starts with donor and recipient populations incapable of triggering gut inflammation before HGT happens. A second factor compromising the efficiency of SopE ϕ -mediated *hilD* transfer may reside in the instability of *hilD*, with rapid accumulation of inactivating mutations (thus out-pacing sluggish phage-mediated *hilD* transfer). As conjugative P2 transfer occurs efficiently without gut inflammation, we therefore switched to the conjugation approach. This is explained in the revised manuscript (lines 78-79).

Figure caption: Low rate of cooperative virulence restoration from cheaters by bacteriophage transfer *in vivo*. The same region of *hild*-cat that is contained within the ρ Vir^{High} plasmid (Fig. S1A) was cloned into SopE ϕ , in the place of the *sopE* gene. The resulting prophage is called ϕ Vir. For bacteriophage transfer experiments, *S.Tm* Δ *hild* SL1344, which naturally contains the SopE ϕ phage, was used as a donor, and ATCC 14028S Δ *hild* was used as a recipient. As a control, SopE ϕ labelled with a kanamycin resistance cassette was used. All strains were ampicillin resistant due to the pM975 plasmid.

A) *In vitro* transfer of ϕ Vir (solid circles; n=6) is as efficient as SopE ϕ (hollow circles; n=6) in the presence of 0.25 μ g/ml mitomycin C. Donors (blue) and recipients (green) were mixed at a 1:1 ratio (500 CFU each) and allowed to grow for 24h in 5 ml LB with mitomycin C. Lysogens were quantified by selective plating. Mann-Whitney U test $p \geq 0.05$ (ns). Dotted line indicates the detection limit. Solid lines indicate medians.

B) 100% of ϕ Vir lysogens from panel A expressed *ttss-1* as determined by a colony blot (representative image shown). This indicates that virulence could be restored *in vitro*.

C) Transfer of ϕ Vir is inefficient *in vivo*. The same Δ *hilD* donor and Δ *hilD* recipient pairs as in panel A were given to ampicillin pre-treated C57BL/6 mice with the ϕ Vir donors in 50-fold excess (25000 CFU by gavage in total; n=3; solid circles). For the SopE ϕ control, wild-type strains were used (i.e., no Δ *hilD* deletion; 1:1 ratio in the inoculum; 500 CFU each; n=1; hollow circles; as in Diard *et al.* 2017, Science, PMID: 28302859). Feces were collected daily for 3 days and populations of donors (blue), recipients (green), and lysogens (red) were enumerated with selective plating. Dotted line indicates the detection limit. Solid lines indicate medians.

D) 100% of ϕ Vir lysogens from panel C expressed *ttss-1* as determined by a colony blot (performed on the mouse with $>10^7$ CFU lysogens / g feces; right-most image). The ϕ Vir construct is unstable since *ttss-1* negative clones emerged in the ϕ Vir donor population by day 3 p.i. (left-most image). A colony blot from the recipient population served as a control that recipients were initially *ttss-1* negative (some *ttss-1* positive lysogens can be observed; middle image). Red arrows give examples of *ttss-1* negative clones; green arrows give examples of *ttss-1* positive clones.

2-The sequencing of clinical isolates in previous studies showed that *hilD*-cheaters emerge frequently in the *Salmonella* *in vivo*-population. In this regard, the authors predicted that HGT can facilitate the reversal of these cheaters to virulent clones. Were there any revertants detected among the clinical isolates? I understand that the authors employed *Salmonella* as a tractable model to make inferences about other systems. However, it will be interesting if there is precedence for virulence restoration in cheaters among the natural isolates of *Salmonella*.

Response: The P2-encoded functional *hilD* allele provided us with a versatile experimental system to probe HGT-mediated emergence and stabilization of cooperative virulence in an *in vivo* setting. However, the *hilD* gene does not naturally occur on a MGE, being a functional phage or a plasmid, which should limit chances of reverting cheats into cooperators. Generalized transduction may be another way to transfer functional *hilD* allele from cooperators to cheats in the *Salmonella* population. Nevertheless, such events would remain unnoticed by comparative genomics. More generally, revertants are difficult to detect in natural isolates and in infection models like ours. As far as we know, a thorough longitudinal pathogen population analysis where the rise and fall of cheats would be carefully monitored has never been performed during non-typhoidal *Salmonellosis* in natural conditions. We have explained this in the revised manuscript (lines 261 To 262).

3- The authors used SPI-2 T3SS- *Salmonella* strains to infect C57BL/6 mice (Nrap-). This is to reduce the inflammation and avoid the rapid death of the host. Why is this a better strategy than infecting 129 SvEv Nrap+ mice with SPI-2 T3SS+ *Salmonella*? The Nrap+ vacuoles in the latter mouse strain should limit *Salmonella* expansion, supporting infection for longer time periods than C57BL/6 without the need to inactivate SPI-2 T3SS. The latter is required by *Salmonella* strains to establish their intravacuolar habitat. This allows the establishment of intracellular reservoirs from which a subset of the bacteria will escape and hyperreplicate in the cytoplasm of epithelial cells (Klein *et al.*, 2017). Given that SPI-1 T3SS was shown to be important for the expansion of the cytosolic *Salmonella* population, this phase of the infection may select for the *hilD*+ co-operative clones. In this regard, the model employed by the authors is potentially lacking some relevant selective forces, which can impact the findings. This needs to be addressed.

Response: In this study, we use Spi-2 mutants to prevent deadly systemic spread of *Salmonella* in C57BL/6J mice. The reviewer is right, it is possible to perform chronic infections by using Nrap+ lines such as 129 SvEv mice and fully virulent *Salmonella* strains. In fact, this was done in a previous publication from our laboratory assessing the impact of antibiotic treatments on the evolution of cooperative virulence (Diard M. *et al.* Current Biology 2014

<https://doi.org/10.1016/j.cub.2014.07.028>). This publication also demonstrates that, in the streptomycin pre-treated model, the intracellular reservoir of virulent clones is only relevant for stabilizing cooperation when the host is treated with a second antibiotic that empty the intestinal lumen and allows growth of re-seeding bacteria from the tissues. In the absence of antibiotic treatment, cheaters outcompete cooperators. Moreover, once cheaters at the level of the plasmid become fixed in the population, rare reseeded events of cells carrying the cooperative allele on MGE (cf. Bakkeren Nature 2019) or on the chromosome should not be able to restore virulence. Therefore, the results obtained with C57BL/6J mice in the current piece suggest that testing hILd transfer in Spi-2 positive *S.Tm* strains in 129 SvEv or other Nramp+ mice would essentially yield similar results. We have explained this in the revised discussion section (lines 267-280).

4-FigS1: The authors showed a correlation between T3SS-1 expression and growth defect. They attributed the fitness cost associated with pVir-high to the overexpression of T3SS-1, mediated by HilD. What about the other HilD-regulated genes, like flagellar motility genes? How much do they contribute to the fitness defect? The authors did not acknowledge this possibility. If this was addressed in previous studies, the authors should mention that.

Response: We do attribute the fitness cost of pVir-high to the over-expression of the HilD regulon which comprises T3SS-1, the flagella, the SPI-4 adhesin, chemotactic receptors and other functions covering more than 250 genes in total (Colgan et al. PloS Genetics 2016; PMID: 27564394). In Sturm et al. PloS Path. 2011 (<https://doi.org/10.1371/journal.ppat.1002143>), the growth defect was attributed in part to SPI-1 encoded translocon and T3SS-1-secreted effectors. A recent preprint shows that additional costs are associated with other co-expressed functions controlled by HilD (Sobota et al. 2021 BioRxiv). We have discussed this in the revised introduction (lines 52-53).

5- The authors infected mice with clonal *Salmonella* populations to mimic the effects of genetic drift during infection transmission. It will also be interesting to investigate the population dynamics during natural transmission (e.g. co-housing abx-treated animals with the infected ones). This will be a better proxy to what occurs in nature.

Response: At first approximation, transmission of *Salmonella* in natural settings should include a substantial population bottleneck due to environmental stress (e.g., desiccation, light, predation, etc) probably leading to near clonal growth in contaminated foods. This is not recapitulated in co-housing experiments where mice often ingest entire fecal pellets right after shedding. This means no bottlenecking, especially when the intestinal niche is emptied by the antibiotic pretreatment suppressing colonization resistance. Moreover, the point of the transmission experiments in this paper was to simulate the most extreme population bottleneck in order to demonstrate that clonal transmission should favor cooperation by maximum assortment. Hence, we had to proceed via controlled oral gavage. This is discussed lines 200-202.

Minor comments:

6- **Fig S3A:** Why are there more transconjugants than recipients at some time points (e.g. day 2)? The authors mentioned that transconjugant enumeration was done after replica plating. Please clarify.

Response: A transconjugant is a recipient (kanamycin resistant, kan) carrying the plasmid and, with it, an extra-resistance to chloramphenicol (Cm). Transconjugant were enumerated on double-selection plates (Cm+Kan) and recipient on single kan selection. Plating allows analyzing up to 200 clones. Therefore, when 99% of recipients are transconjugants, we count 2 KanR/CmS clones for 198 KanR/CmR clones, i.e., more transconjugants than Recipients without plasmid. The detection limit was a 2-log difference between transconjugants and the remaining fraction of recipient cells. This is now explained in more detail in the experimental procedures (lines 504-512).

7- **Fig 1F:** This figure doesn't add new information, so please remove it or move to the

supplementary data. The trends are clear in the figures preceding this one.

Response: We agree. The panel 1F is presented in supplementary figure S3D of the revised manuscript.

8- **Fig 3D-F:** The information in these figures can be inferred from the other data. This sort of redundancy is distracting to the reader. Also, the positive correlations presented in 3E and 3F are weak despite being significantly different from zero.

Response: We agree that Fig. 3D-F might be distracting in the main figures. However, as these plots highlight correlations between presence of *ttss-1* expressing cells, inflammation and total population size of *Salmonella* at the end of the infection, which, we think may be useful to convey our message to some readers, we decided to transfer these plots to supplementary fig S5.

9- **Lines 61:** The authors used “cooperative virulence” to refer to the co-expression of *hilD* in trans and T3SS-1 in cis. However, the reader can confuse this with lineage co-existence (e.g. the co-existence of the virulent *hilD*+ clones and the *hilD*- cheaters). This needs to be clearly explained in the manuscript to make it easy for the reader to follow the narrative.

Response: “Cooperative virulence” refers to virulence as a cooperative trait (i.e., cheatable) in a general. This is now explained in the introduction (line 65).

10- **Line 125:** How many evolved isolates were sequenced?

Response: Sequenced clones are listed in tables S1 to S4. 11 clones carrying pVir low and 14 clones carrying pVir High were sequenced to detect mutations in both chromosome and pVir plasmids. The sequencing results match the colony blot readout for expression of the *HilD* regulon using *SipC* expression as a proxy. Loss of *SipC* expression correlates with mutations in *hilD*. This number of clones is now specified on line 130.

11- **Line 128:** Please indicate that ‘MGE’ stands for ‘mobile genetic elements’

Response: This is now corrected.

12- **Line 128:** The authors described the slower loss of T3SS-1+ clones in the pVir-low population compared to pVir-high counterpart as ‘striking’. Isn’t that expected given that the fitness cost of expressing *HilD* is higher in the latter?

Response: This is now corrected.

13- **Line 136:** Change “with” to “by”

Response: Corrected.

14- **Lines 175-176:** “we concluded that the proportion of cooperators do contribute to disease transmission.” Change this sentence to “.....contribute to disease development post transmission” because transmission in this experiment was artificial. Thus, the emphasis should be on how the proportion of cooperators affects inflammation development. Any claims regarding the efficiency of pathogen transmission between hosts are speculative.

Response: Agreed and corrected.

15- **Line 177:** What is the “sufficient” proportion of cooperative clones required in the population to trigger inflammation? This requires testing different proportions of the cooperative lineages.

Response: This refers to data from Diard M. et al. *Current Biology* 2014 showing that a population made of 99% cheaters cannot trigger the disease after transmission. Moreover, the Supplementary figure 1 from Ackerman M. et al. *Nature* 2008 provides a fair approximation of the amount of cells expressing T3SS-1 that is necessary to trigger inflammation (between 10 and 50%). We have amended the results section to point this out to the reader (**lines 182-187**).

16- **Lines 278-280:** The authors mentioned anti-virulence compounds, but then suggested the administration of cheaters to destabilize the *Salmonella* population. The latter is a form of biocontrol; hence, this part is confusing. A discussion of compounds that can potentially accelerate the emergence of cheater/avirulent clones will be more appropriate.

Response: We do not know of compounds accelerating the fixation of cheaters, but we are working on it. This part of the discussion has been reoriented toward the biocontrol approach.

17- **Line 455:** Please indicate which mouse strain was used, C75BL/6 J or N.

Response: C57BL/6J. Corrected.

Reviewers' Comments:

Reviewer #2:

Remarks to the Author:

I have no further concerns.